# Prime Editing: An All-Rounder for Genome Editing

**DOI:** 10.3390/ijms23179862

**Published:** 2022-08-30

**Authors:** Chenyu Lu, Jingyu Kuang, Tong Shao, Sisi Xie, Ming Li, Lingyun Zhu, Lvyun Zhu

**Affiliations:** Department of Biology and Chemistry, College of Sciences, National University of Defense Technology, Changsha 410073, China

**Keywords:** prime editing, genome editing, technical optimization, applications

## Abstract

Prime editing (PE), as a “search-and-replace” genome editing technology, has shown the attractive potential of versatile genome editing ability, which is, in principle, currently superior to other well-established genome-editing technologies in the all-in-one operation scope. However, essential technological solutions of PE technology, such as the improvement of genome editing efficiency, the inhibition of potential off-targets and intended edits accounting for unexpected side-effects, and the development of effective delivery systems, are necessary to broaden its application. Since the advent of PE, many optimizations have been performed on PE systems to improve their performance, resulting in bright prospects for application in many fields. This review briefly discusses the development of PE technology, including its functional principle, noteworthy barriers restraining its application, current efforts in technical optimization, and its application directions and potential risks. This review may provide a concise and informative insight into the burgeoning field of PE, highlight the exciting prospects for this powerful tool, and provide clues for questions that may propel the field forward.

## 1. Introduction

The realization of gene manipulation has allowed rapid progress in biology in recent years. Unlike early genetic engineering techniques, which can only randomly insert exogenous or endogenous DNA into the host genome, precise genome editing technologies enable the specific editing of target genes, and it has been widely applied in genetic research, gene therapy, and genetic improvement.

Scientists have found that some nucleases can be modified to identify specific sites within the genome and create double-strand DNA breaks (DSBs) to induce DNA damage repair, thus enabling the desired editing of the target genome. Genome editing techniques using nucleases include zinc finger nuclease (ZFN), transcription activator-like effector nuclease (TALEN), and CRISPR-Cas9 system [1]. ZFN and TALEN have highly repetitive peptide motifs with complex protein structures, which make them time-consuming and costly in terms of preparation and synthesis [2]. Furthermore, these techniques suffer from several problems, such as high difficulty in expression, susceptibility, and difficulty in protein spatial folding, when these motifs work in cells. In contrast, the CRISPR-Cas9 system relies on multiple sgRNAs for parallel locus editing, making it the most rapid and cost-effective method of gene editing [3]. Conventional CRISPR-Cas9 systems have expanded their repertoire of editing and perturbation modalities, including knockdown, activation, and repression of gene expression. Moreover, to install or remove specific sequences or bases from the target genome, the DSB introduced by CRISPR-Cas9 is parallel to homology-directed repair (HDR), resulting from the presence of the exogenous DNA template. However, this method is often inefficient in mammalian cells and prone to unintended mutations [4,5]. Through the incorporation of different deaminases with the CRISPR-Cas system, base tabliediting (BE) has been developed to overcome these deficiencies. Base editing enables four transition mutations (i.e., C→T, G→A, A→G and T→C) without introducing DSBs, with very high efficiency (even up to 100% in cultured mammalian cells), and a low off-target rate [6,7,8]. An exciting advancement could be achieved if the utility and versatility of genome editing was enhanced by enabling all types of edits. In 2019, Anzalone et al. coined prime editing (PE), a significant innovation in genome editing technology that can perform all 12 possible base-to-base conversions, insertions, and deletions [9]. PE has greater targeting flexibility, no bystander products and a broader editing range than BE, highlighting the critical addition of programmable genome editing technologies [9,10]. Although the editing efficiency of PE is not yet comparable to that of BE, with continuous efforts, PE editing efficiency is now up to 53.2% and its unique advantages make it promising for a wide range of applications [11]. The expansion of the CRISRP-Cas9 genome toolkit has promoted the extensive application of this cutting-edge technology, particularly in clinical medicine.

## 2. Origin of the PE System

The conventional CRISPR-Cas9 system consists of two main components: a Cas9 protein that performs a DNA double-stranded cleavage function, and a single guide RNA (sgRNA) containing a sequence complementary to the target DNA sequence. This Cas9-sgRNA complex binds to the target DNA by recognizing the protospacer-adjacent motif (PAM) sequence and then generates a DSB, which prompts cells to perform DNA repair. Precise editing of the target sequence is achieved by HDR, while another mechanism, non-homologous end joining (NHEJ), randomly introduces insertions or deletions (indels) into the edited DNA sequence [12].

In addition to its nuclease activity, Cas9 can recruit proteins and RNAs to target DNA sites, and it thus has a strong potential to become a powerful tool for regulating sequence-specific genes. Mutations in the two nuclease domains of Cas9 result in dead Cas9 (dCas9), which retains the ability to bind DNA guided by RNA but loses its nuclease activity such that it cannot cut DNA strands [13]. The transcriptional repressor or activation domain is fused with dCas9, and the complex can efficiently interfere with or activate gene expression; this is called CRISPR interference (CRISPRi) or CRISPR activation (CRISPRa) [14]. However, these techniques are ineffective in correcting point mutations in genetic diseases; therefore, BE technology was spawned. Liu et al. adopted the CRISPR-Cas9 system by fusing dCas9 with different types of deaminases, allowing for four transition mutations (i.e., C→T, G→A, A→G and T→C) without producing DSBs [6,15]. However, BE can only perform two transversion mutations (C→G and C→A); for example, the T∙A-to-A∙T mutation in the sixth codon of the HBB gene, which is common in sickle cell disease, cannot be corrected by BE [6,16]. In addition to this deficiency, BE can lead to bystander editing, resulting in undesired byproducts [17]. To fill these gaps, PE, a state-of-the-art genome editing tool, was developed.

## 3. Principle of the PE System

The prime editor is composed of two parts (Figure 1): a reverse transcriptase (RT) that incorporates a Cas9 nickase (H840A mutation), which can only produce single-strand breaks, and a 3′-extended sgRNA containing a primer binding site (PBS) and a RT template and is referred to as pegRNA. When the PE system binds to the target DNA through the guidance of pegRNA, Cas9 nickase cleaves the strand containing the PAM, exposes its 3′ end hydroxyl group, and binds to the PBS on the pegRNA, and the prime reverse transcription of the new DNA containing the target edit is performed according to the RT template. At the end of this step, the edited DNA strand has two redundant overhangs: one unmodified 5′ flap and one purposed 3′ flap. In this case, the 3′ flap can be retained and the other can be discarded. Later, the overhang containing the edit integrated into the desired site through a cellular DNA repair system [9,18].

The PE system can mediate the full range of point mutations, insertions, and deletions of small fragments and has greater targeting flexibility than BE in the distance between the PAM and the editing site without resulting in bystander editing. These advances have rendered PE a universally precise genome editing tool.

## 4. Improvements in Prime Editors

Although PE has overcome the limitations of previous gene editing technologies, its practical application has many shortcomings. First, the available genomic scope that the PE system can target is constrained by the PAM-sequence-dependent recognition of Cas9 nickase; second, the editing efficiency of the PE system varies unpredictably among different cell types due to the differences in target sites and editing types, and the performance of PE in many organisms is still unknown; and third, the original PE enzyme is too large to be loaded using traditional delivery tools. To make the PE system truly effective in broad applications, efforts have been made to address many of the most noteworthy barriers (Figure 2).

### 4.1. Relaxed PAM Restriction for PE

Before proceeding to clinical treatment, improving the PAM coverage of PE systems is essential for installing genome-wide editing for all loci. Extending the editing range of PE systems depends mainly on Cas9 enzymes that can recognize PAM. The current PE system relies heavily on SpCas9 enzyme, which requires NGG PAM sequences that occur on average once in every 16 randomly selected genomic loci, greatly limiting the range of targets for gene editing [19,20]. Thus, researchers have long worked to discover and engineer upgraded Cas9 enzymes to expand their compatibility with various PAMs. In 2015, Keith lab developed two SpCas9 variants, SpCas9-VRQR and SpCas9-VRER, which recognize NGA and NGCG, respectively [21]. Three years later, Hu et al. used the phage-assisted continuous evolution (PACE) strategy to construct xCas9-3.7 variants that recognize the PAMs of NG, GAA and GAT [20]. In the same year, Nishmasu et al. reported NG-recognizable SpCas9-NG variants [22]. Furthermore, similar to PACE, Liu et al. developed three newly evolved SpCas9 variants, expanding the targetable PAM to NRNH (H: A/C/T) [23]. Immediately afterward, Russell et al. upgraded SpCas9 strongly to obtain the SpRY variant recognizable PAM covering NRN and NYN (Y: C/T), almost completely free from the limitation of PAM, which is currently the most compatible SpCas9 variant for PAM [24]. Based on the evolution of the Cas9 variants described above, these variants have been modified to obtain the corresponding nickase, some of which have been used in PE systems to considerably extend the editing range; however, extensive experiments are still needed to verify the effectiveness of PE using these optimized variants for genome editing [25].

### 4.2. Improvement of the Editing Efficiency

Researchers are currently most concerned about the editing efficiency of PE, which is criticized for its low efficiency at specific loci or certain types of editing compared with BE [18,26]. According to a seminal article reporting PE, scientists have been trying to improve its efficiency. The primitive PE system, PE1, was developed by fusing Moloney murine leukemia virus (M-MLV) RT to the C-terminus of Cas9 (H840A) nickase. The pegRNA used was an extension of either end of sgRNA with a PBS sequence (5–6-nt) and an RT template (7–22-nt) [9]. It achieved a maximum editing efficiency of 0.7–5.5% for introducing transversion point mutations and 4–17% for targeted insertions and deletions at different loci. Building on PE1, Liu et al. proposed the idea of improving the efficiency of prime editing by modifying RT. They tried multiple mutants of M-MLV RT and eventually found that RT with the introduction of M3(D200N, L603W, and T330P), T306K and W31S worked best, resulting in a PE system called PE2. Compared to PE1, PE2 increases the efficiency of harboring point mutations by 1.6- to 5.1-fold on average, exhibits a higher editing efficiency in indels, and is compatible with short PBS, according to the testing of various RT mutations. Furthermore, Liu et al. investigated the relationship between the pegRNA structure and the efficiency of PE2 and, on the basis of the experimental results, recommended optimizing the PBS length starting from 13-nt, RT template length starting from 10 to 16-nt, and designing the pegRNA 3′ extension without the first base of C [9]. Moreover, they were inspired by their former work on optimizing the BE design to propose the strategy of adding a sgRNA to make a non-edited strand break. The newly obtained prime editor is called PE3, and it has approximately three times the editing efficiency of PE2 but introduces a higher level of indels.

Inspired by the secondary structure of pegRNA, Wang et al. proposed that the PBS at the 3′ end of pegRNA is complementary to the part of the spacer at the 5′ end, which may lead to cyclization of pegRNA during annealing and thus hamper the function of PE [27]. To prevent pegRNA cyclization, they fused a 20-nt Csy4 recognition site to the 3′ end of pegRNA, which can form a hairpin structure, and fused Csy4-T2A (Csy RNase) with Cas9 nickase and the RT complex to obtain a PE3-based enhanced PE system (ePE). Their experiments verified that ePE improves the efficiency of target editing compared to the PE system using canonical pegRNA but results in more indels.

Liu et al. also developed the idea of modifying pegRNA [28]. Their study pointed out that the 3′ extension of pegRNA is susceptible to exonucleolytic degradation when exposed to cells, poisoning the activity of prime editors and thus reducing editing efficiency. This finding suggested that the stability of pegRNA affects the editing efficiency of the PE system. Then, they came up with a solution of adding structured RNA motifs to the 3′ end of pegRNA to improve the stability of pegRNA and prevent degradation of the 3′ extension. A minimal class of naturally obtained RNA structural sequences with a well-defined tertiary structure, that is, a modified prequeosine1-1 riboswitch aptamer (evopreQ1, 42-nt in length) was selected because it can be easily obtained through chemical synthesis [29,30]. The pegRNAs obtained by adding this pseudoknot are called epegRNAs. Moreover, to reduce the possibility of the aptamer interfering with pegRNA function during prime editing, an 8-nt linker was attached between the 3′ end of the PBS of epegRNA and evopreQ1. The results showed that optimization of pegRNA resulted in a 3–4-fold increase in efficiency in several cell lines (HeLa, U2OS, K562, and primary human fibroblasts) with no increase in off-target effects.

Similarly, Li et al. stabilized pegRNA by inserting a C/G pair or changing each non-C/G pair to a C/G pair in the small hairpin of pegRNA(apegRNA), resulting in an average 2.77-fold improvement in indel-editing efficiency when applied it to PE3 [31]. They also adopted the second strategy: introducing same-sense mutations at appropriate positions in the RT template portion of pegRNA (spegRNA), which increased the base-editing efficiency by an average of 353-fold. Zhang et al. resisted pegRNA degradation by appending a viral exoribonuclease-resistant RNA motif (xrRNA) to the 3′ end extension of pegRNA, and the resulting xrPE had 3.1-, 2.5-, and 4.5-fold enhancements, respectively, for base conversions, small insertions, and small deletions in a variety of cell lines compared to PE3 [32]. Recent studies have also reported the modification of pegRNA with a human telomerase RNA (hTR) G-quadruplex (G_3_N_1-3_G_3_N_1-3_G_3_N_1-3_G_3_) to improve the stability of pegRNA, resulting in G-PE with editing efficiency similar to that of the PE system using evopreQ1, while the G-quadruplex is shorter than evopreQ1 and xrRNA [33].

In addition to the above-mentioned improvements in the pegRNA structure, David Liu’s group explored the intracellular determinants that are particularly related to DNA mismatch repair (MMR), which may hinder PE function and cause unintended indels, using Repair-Seq, a high-throughput pooled genetic screening approach that can measure the effects of various loss-of-function genomic perturbations on the outcomes of PE function [34,35]. Inspired by the finding that cellular pretreatment with siRNAs targeting MMR enhances the editing efficiency of PE, a strategy for the simultaneous delivery of prime editors and MMR repressor units was proposed. After continuous attempts and improvements, the transient expression of the MMR repressor protein MLH1d in the PE2 and PE3 systems, which are called PE4 and PE5, respectively, significantly enhanced editing efficiency. Moreover, the study also engineered the PE2 protein and found a new PE structure (PEmax) that uses a human codon-optimized RT, a 34-aa linker that includes a bipartite SV40 NLS, an added C-terminal c-myc NLS, and R221K N394K mutations in SpCas9 [36,37,38]. Furthermore, the combination of the two enhancements produced PE4max and PE5max, which were then used together with the previous epegRNA to significantly improve the performance of PE [28]. Among the different combinations of these optimizations, PE5max with epegRNA exhibited the best performance and is recommended for most applications with PE. Gao’s group adopted a double pegRNA strategy that dramatically improved PE editing efficiency by using two trans-pegRNAs to simultaneously edit both strands of DNA to achieve the same mutation [39]. They then modified the RT enzyme structure to improve enzyme activity and DNA synthesis efficiency, which they believed was related to the PE efficiency [40]. They tried two strategies: one was to remove the RNase H domain of RT and the other was to incorporate a viral nucleocapsid protein with nucleic acid chaperone activity. Both strategies independently increased the editing efficiency of prime editing in plant cells by approximately 1.8 to 3.4-fold. The two modifications were then used in combination to increase the editing efficiency by an average of 5.8-fold compared to the original plant PE system (see Section 5.2 Agriculture).

After the above refinement (Table 1), the highest editing efficiency that can be achieved by the PE system reached approximately 50%, which is a 10-fold improvement compared to the first generation of PE system [10]. However, PE is outperformed by BE in most of the mutations that BE can realize. Editing efficiency remains a significant challenge for PE in the future. Further pegRNA engineering should remain the main method to improve PE editing efficiency in the future. A more in-depth study of the mechanism of PE can also help provide new ideas for improving editing efficiency.

### 4.3. Delivery of PE Agents

Apart from editing efficiency, successful genome editing depends on the efficiency of the editing tools that can be delivered to the target. The large size of genome editing machines and the need to simultaneously deliver multiple macromolecules into cells, tissues, or organs pose a significant challenge. There are wo main types of delivery methods for genome-editing systems: virus-based delivery and non-viral delivery. A range of viral vectors have been used to deliver genome editing tools, including adeno-associated viral (AAV), lentiviral, retroviral, adenoviral, and baculoviral vectors. AAV and lentiviral vectors have been used in most studies [41].

The AAV vector has clinically advantageous features, such as replication incompetence, low immunogenicity, different serotypes, and tissue specificity [41,42]. Two types of AAV vectors have been approved by the US Food and Drug Administration (FDA) for use in the treatment of retinal degenerative disease and spinal muscular atrophy type 1 [43]. The AAV vector can only pack up a gene cargo within ~4.7 kb [44]. The prime editor includes dCas9 (~4.2 kb), a reverse transcriptase domain (~2.0 kb), promoters, and other essential components. Thus, the length of the prime editor clearly exceeds the packaging capacity of the AAV vector. The prime editors can be split such that the size of each part is within the vector capacity. The researchers used a dual AAV approach: the prime editor N-terminal and C-terminal expression cassettes were delivered in two separate AAV vectors and then delivered into the adult mouse retina, which also achieved editing of the target site [45,46,47]. However, split-PE may have lower efficiency in vivo than intact PE because of imperfect construction of the spatial structure [48,49]. In addition, the choice of split sites for the split-PE system is limited. To overcome this limitation, Zheng et al. used truncated RT to further reduce the size of PE and improve intracellular safety [50]. They developed a compact PE that did not contain the RNase H domain of RT, increasing flexibility in the selection of split sites, while maintaining editing efficiency.

To meet the requirement of AAV vector-based delivery for the full-length prime editor, reducing the size of Cas9 nickase could be an alternative optimization strategy. Several common and current miniaturized Cas9 variants are listed in Table 2, predicting the size of prime editors composed of these variants and showing which variants constitute the PE systems expected to be delivered by AAV vectors [51,52,53,54,55,56,57,58,59]. SaCas9 was engineered to construct adenine base editors (ABEs), allowing Cas9-ABEs to be delivered by AAV vectors with increased on-target DNA editing [60]. Meanwhile, experiments on the modification of Cas9 enzymes are being conducted, but optimized Cas9 variants have not been used in the PE system. Further attempts should be made to observe the outcomes of PE incorporation with different Cas9 variants.

In addition to the AAV vector, another commonly used delivery method, the lentiviral vector (LVV), can theoretically carry ~8 kb of insert fragments composed of a ~2 kb intrinsic sequence carrying a genomic RNA and a chromosomal integration constitute and a ~6 kb programmable fragment, making the delivery of PE systems with LVVs a common choice for many researchers performing cellular experiments in vivo [61,62]. Nevertheless, considering biosafety issues, the potential for insertion of target exogenous genes into the host genome by LVVs makes their clinical application controversial, and doctors take a conservative approach to this delivery method [63]. Moreover, LVV can target a more limited number of cell types, has less tissue specificity, and is less efficient in vivo than the AAV vector [64,65,66]. In addition to the two viral vectors mentioned above, the recently developed retrovirus-like protein PEG10 packages and engineered DNA-free virus-like particles, which can further reduce immunogenicity, are also expected to be used for the delivery of PE systems [67,68].

Non-viral delivery has the advantage of being suitable for the transient expression of genome editing tools with low DNA toxicity [41,69]. Most importantly, non-viral platforms are expected to pack PE agents because they can be delivered without packing capacity limitations [70]. Cas9 ribonucleoprotein (RNP), which is a popular non-viral platform, has been directly adapted for delivering PE. Yeh et al. successfully achieved wide-scale editing of target sites by delivering PE RNP directly into zebrafish embryos, which may further improve the editing efficiency of PE because RNP minimizes the time of exposure to the editing unit [71]. Other forms of non-viral delivery, such as lipid nanoparticles and cell-penetrating peptides, offer the advantages of low immunogenicity and broad targeting range and have potential applications in the delivery of PE systems [72].

### 4.4. Algorism Development for the Design of pegRNA

During the exploration of the diversity of PE applicability in species, tools for PE system design and the prediction of editing effects were merged to facilitate the experiments. pegRNA design tools, such as pegFinder, PE-Designer, Prime Induced Nucleotide Engineering Creator of New Edits, PrimeDesign, and PnB Designer, require only the wildtype or reference DNA sequence of the target site and the edited/desired DNA sequence and the sequence and related detailed information of the pegRNA and sgRNA(if using PE3) that meet the requirements (specific length of RT, PBS, etc.) [73,74,75,76,77]. Several algorithmic models are debugged to select the target sequence that obtains the highest efficiency based on the predicted results. To quickly obtain a picture of the editing of the target region, Hwang et al. invented a PE-Analyzer, which can easily analyze the PE high-throughput sequencing results of cells that have undergone PE, providing information, such as the distribution and specific values of each type of mutation [74]. To demonstrate the utility of these tools, developers also implemented gene mutations in human cells using software-designed pegRNAs and ngRNAs [73,75,76]. Studies of these tools have facilitated PE experiments and inspired new perspectives for prime editor design improvements.

Many elaborate efforts have been made to achieve substantial progress in the utilization of PE systems in clinics and research. However, more comprehensive and delicate optimizations are necessary because current strategies often account for only one aspect of the difficulties encountered by the PE system. For example, split-PEs are available for AAV packaging but underperform full-length PE in terms of efficacy. Meanwhile, this progress has inspired researchers to take a multi-faceted approach to enhance PE, which can be unified with the powerful CRISPR screening technology to explore the factors affecting PE performance at a deeper and a broader scale and thus improve the prime editors.

## 5. Applications of PE

Given the revolutionary potential of PE in life sciences as a versatile genetic modifier, it is now an exciting time for researchers to test its transformative applications in various fields, such as medicine, agriculture, and other biotechnologies in various species (Figure 3 and Table 3).

### 5.1. Medicine

The leading application field of the PE system is clinical medicine, which is currently expected to be employed mainly for the treatment of genetic diseases. Anzalone et al. validated in HEK293T cells that PE can correct the major mutations that cause sickle cell disease (SCD) and Tay-Sachs disease [9]. The first generation of targeted base conversion mutations in animals with the PE system was demonstrated by Smith et al., validating the efficacy of PE in mice [78]. According to Schene, PE can achieve the functional repair of intestinal organoids with DGAT1 defects and liver organoids with Wilson disease (ATP7B), demonstrating that PE can be applied to precisely edit not only two-dimensional growth cell lines but also three-dimensional organoids [79]. Jang et al. utilized PE to successfully repair disease-causing mutations in mice with genetic liver disease, hereditary tyrosinemia, genetic eye disease, and Leber congenital amaurosis [80]. These studies raise the possibility of using PE as a therapy for multiple diseases. Meanwhile, many studies have applied PE in various single-base editing, insertion, and deletion attempts on a variety of mammalian cells and organisms, generating models of genetic mutation diseases and providing a convenient tool for mechanistic studies and drug screening [26,81,82]. Considering the huge potential of PE in future therapeutic applications, David Liu and Andrew Anzalone recently founded Prime Medicine, which specializes in the commercial development of PE technology for gene therapy, and the company raised $315 million within a short period of time. This initiative is expected to accelerate the application of PE in clinical settings.

### 5.2. Agriculture

Given that genome editing is one of the most important strategies in plant biology, and PE is rising in popularity among gene editing tools, many researchers expect to apply PE in plants to improve yield and insect and disease resistance. Recently, several groups have independently developed plant PE systems in rice, which are representative of monocotyledonous plants with high economic value [83,84,85,86,87,88]. According to the data they obtained, although various versions of PE agents have been tested and some measures have been taken to optimize the codon, promoter, or select Cas9 and RT variants, PE editing in rice currently exhibits low efficiency.

Attempts on other species are also underway. Lin et al. established plant PE (PPE) for wheat to perform diverse edits at target sites [89]. Jiang et al. pioneered the editing of two non-allelic targets with prime editors in maize, and confirmed the hypothesis that enhanced pegRNA expression could improve editing efficiency [11]. Lu et al. employed PE in tomato, a representative dicotyledonous plant, to achieve precise genome modifications [90]. These studies raise the possibility that PE systems have broad application prospects in the plant world. However, the relatively low editing efficiency of PE in plants poses a major barrier for plant biotechnology and agricultural applications. Currently, the highest editing efficiency achieved in plants using the PE system is 53.2%, which was obtained when the W542L and S621I double mutations were respectively generated in ZmALS1 and ZmALS2 in maize [11]. However, on average, the editing efficiency of PE on average in plants is relatively lower than in human cells (20%–50%) [91]. Therefore, efforts should be made to improve the performance of PE in plants through rational strategies, including protein structural evolution, pegRNA optimization, system redesign, and/or cellular determinant screening. The differences in cellular structure and genome composition between plants and animals also account for the required suitability test in plants before applying emerging advances in PE from animal models. Nevertheless, the versatility of PPE makes it potentially advanced for the domestication of wild plants, plant breeding, drug plant production, etc.

### 5.3. Biotechnological Applications in Different Species

In addition to mammalian and higher plant models, PE also offers desired biotechnological applications in various organisms, such as lower vertebrates and micro-organisms. Justin et al. experimented with PE for the first time in a non-mammal (i.e., Drosophila) and reported the installation of PE in the fly germline [92]. Karl introduced somatic mutations in zebrafish embryos with a 30% editing frequency and reproductive transfer [71]. Qian et al. subsequently reported the use of the PE system in rabbits to efficiently and precisely produce the Tay-Sachs disease model [93,94]. Attempts have been made in the field of microbiology. Tong et al. successfully performed large fragment deletions (up to 97 bp) and insertions (up to 33 bp) in Escherichia coli, revealing the industrial application potential of PE in microbial cell factories [95]. The successful employment of PE for precision editing among different species is encouraging and reveals that PE can be applied to generate purposeful editing in a wide range of organisms; however, such interspecies compatibility requires urgent verification through experiments.

## 6. Conclusions

PE technology has been demonstrated to be a powerful and versatile genomic manipulator in terms of precision, attracting tremendous attention for its development. However, like any new tool in its early stage of development, PE has various problems. Extensive efforts have been made to address these problems and promote transformative technologies for good use. Notably, regardless of the application scenario, safety is the main concern, particularly in clinical use, and it has garnered the most attention worldwide. Off-target effects are a common pitfall of gene editing, and PE is no exception [96,97]. Liu et al. showed that prime editing results in far less off-target editing than Cas9 at known Cas9 off-target sites because PE requires two additional hybridization events: the complementation of the target DNA with the PBS region of the pegRNA to initiate reverse transcription and the complementation of the target DNA with the RT product for flap resolution. Many subsequent studies have demonstrated that the off-target rate of PE is relatively low in different species and under various system designs [9,87,98,99]. It should be noted that indels still happen in prime editing, but the frequency is less than 1% in most cases. However, when using PE3, the indel frequency is generally less than 10% [9]. However, a small off-target possibility does not ensure that deleterious mutations can be avoided, and their occurrence leads to unpredictable and even fatal consequences. Although some experimental data have shown that the frequency of off-target editing is very low or even undetectable [9,89,98], this is likely due to the low expression of prime editors in vivo or site-specificity [99]. In particular, the potential of off-target editing to induce oncogenic mutations remains an ongoing problem in the clinical approval of PE-based medicine or other applications. Further high-throughput investigations of whole genomic mutations caused by PE editing should be carried out in different organisms, developing phases, and/or physiological states to verify real off-target effect and prove the reliable clinical relevance of this tool. Nevertheless, the feasibility of PE as a potent tool capable of editing in numerous species suggests its popularity in future research, and the early application of PE in genomic therapy for the benefit of humanity is something to look forward to.

## Figures and Tables

**Figure 1 ijms-23-09862-f001:**
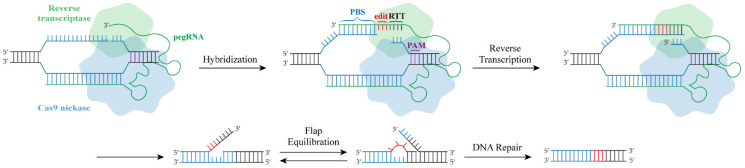
Principle of the PE system. dCas9 binds the target DNA under the guidance of pegRNA and nicks the PAM-containing strand. The hybridization of the exposed 3′ end to the PBS primes the reverse transcription of the nicked DNA strand for the desired edit. Given the preference for the 3′ flap, the 5′ flap cleavages and DNA repair installs the desired edit.

**Figure 2 ijms-23-09862-f002:**
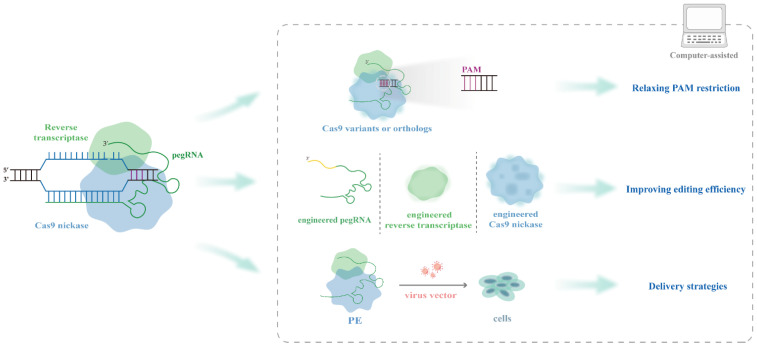
Optimization of PE. Showing the several aspects of optimizing PE and computing assists the development of PE in many ways.

**Figure 3 ijms-23-09862-f003:**
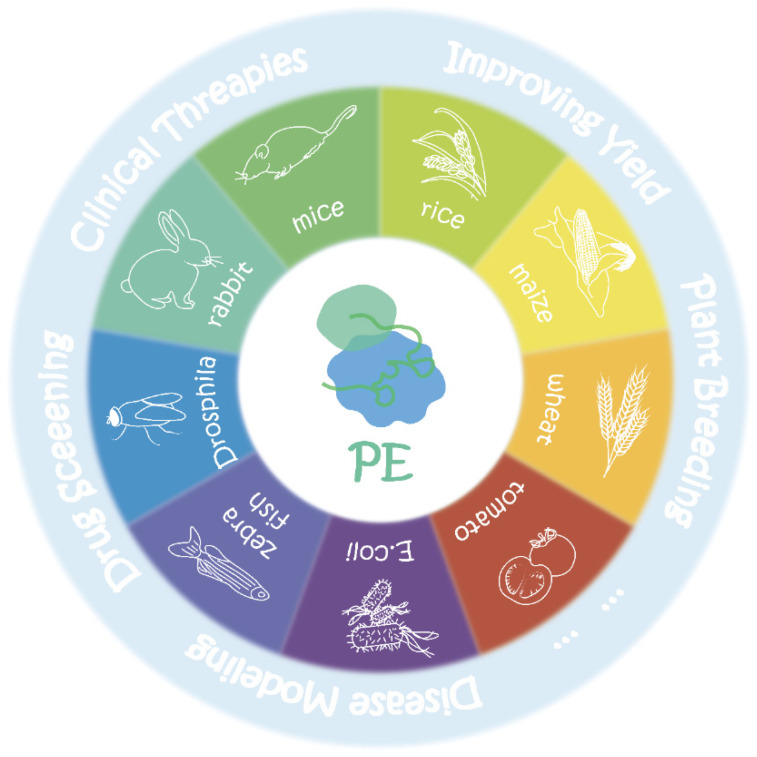
Applications of PE in different species. Current transformative application attempts of PE in various fields, such as clinical medicine, agriculture, and so on.

**Table 1 ijms-23-09862-t001:** Summarization for the refinement in PE.

Designation of PE Systems	Components of PE			Efficiency
	Cas9 Nickcase	Reverse Transcriptase	pegRNA	
PE1	Cas9(H840A) nickase	M-MLV RT	the original pegRNA	0.2–17%
PE2		M-MLV RT (D200N/L603W/ T330P/T306K/W31S)		1.6–5.1-fold
PE3			+ a sgRNA for nicking the non-edited strand	3-fold compared to PE2
ePE	fusing Csy-T2A		+ a sgRNA & fusing a Csy4 recognition site into the 3′ end	1.9-fold *
unnamed	Cas9(H840A) nickase		epegRNA: incorporating evopreQ1	3–4-fold *
aPE			apegRNA: inserting a C/G pair or changing each non-C/G pair to a C/G pair	2.77-fold in indel-editing *
sPE			spegRNA: introducing same-sense mutations	353-fold in base-editing *
x rPE			xr-pegRNA: appending a viral exoribonuclease-resistant RNA motif	3.1-fold in base conversion *
G-PE			incorporating a hTR G-quadruplex	similar to using epegRNA
PE4/PE5	same as PE2 but transient expressing MLH1d		epegRNA	7.7-fold compared to PE2/ 2.0-fold *
PE4max/PE5max	based on PE4/PE5, using a human codon-optimized RT/adding a linker			higher than PE2/PE3
unnamed	based on PPE (similar to PE2)		PBS with a melting temperature of 30℃ and using dual pegRNAs	at least 2.9-fold compared to PPE
ePPE	based on PPE but removing RT’s RNase H domain and incorporating a viral nucleocapsid protein		the original pegRNA	5.8-fold compared to PPE

+: adding to PE1; *: compared to PE3.

**Table 2 ijms-23-09862-t002:** Prediction of PE size with various Cas enzymes.

Ca9 Variants & Others	Cas Size (kb)	Predicted PE Size (kb)	AAV Delivery	Reference
SpCas9	4.2	6.2		[10]
St1Cas9	3.3	5.3		[23]
SaCas9	3.2	5.2	√	[6]
SauriCas9	3.1	5.3		[18]
NmeCas9	3.2	5.3		[16]
CjCas9	3.0	5.0	√	[17]
CdCas9	3.3	5.3		[19]
CasФ	2.1~2.4	4.1~4.4	√	[20]
Cas12f	1.2~1.8	3.2~3.8	√	[21]
CasX	<3.0	<5.0	√	[22]

The prediction is based on the size of Cas9 variants and orthologs, the common RT and pegRNA in Anzalone et al. [8].

**Table 3 ijms-23-09862-t003:** An overview of some representative PE applications.

	Fields	Targets	Details	References
**Applications** **of** **PE**	**Medicine**	HEK293T cells	correcting the mutant HBB allele to wild-type HBB, T•A to A•T (Sickle cell disease)	[9]
			deleting a 4-bp insertion in HEXA (Tay-Sachs disease)	
		mouse neuro-2a (N2a) cells/mouse embryos	generating base conversion in *Ar* gene and *Hoxd13* gene	[78]
		liver- and intestine-derived organoid cells/HEK293T cells/Caco-2 cells	promoting a biallelic 3-bp deletion in DGAT1, creating in-frame deletions in CTNNB1 (liver cancer) and repairing a 1-bp duplication in ATP7B (Wilson disease)	[79]
		Fah^mut/mut^ mice	rescuing a homozygous G-to-A point mutation in *Fah* gene (hereditary tyrosinemia type 1) and correcting a C-to-T transition in *RPE65* gene (Leber congenital amaurosis)	[80]
		mouse N2a cells	installing a G-deletion mutation in *Crygc* gene (cataract disorder)	[81]
		human induced pluripotent stem cells (iPSC)	repairing a missense mutation in *SAMHD1* gene (Aicardi–Goutières syndrome)	[82]
		Fah^−/−^ mouse primary hepatocytes	correcting a Fah mutation (Hereditary tyrosinemia type 1)	[26]
	**Agriculture**	rice (*Oryza sativa*)	editing the *OsALS* (herbicide resistance), *OsIPA* (rice yield) and *OsTB1* gene (lateral braching)	[86]
		the Japonica rice (*Oryza sativa*) variety Zhonghua11 and the winter wheat variety Kenong199 protoplasts	producing a wide variety of edits at genomic sites (including C-to-T, G-to-T, A-to-G, G-to-A, T-to-A, and C-to-A substitutions in rice; including A-to-T, C-to-G, G-to-C, T-to-G, and C-to-A substitutions in wheat)	[89]
		tomato	*GAI*, *ALS2* and *PDS1*	[90]
		maize	introducing W542L and S621I double mutations in ZmALS1 and ZmALS2 (herbicide resistance)	[11]

The content in brackets complements the disease or its function associated with the gene.

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
