# Peer review of "Prime Editing: An All-Rounder for Genome Editing"

_ijms, 2022, doi:10.3390/ijms23179862_

Round 1

Reviewer 1 Report

The manuscript “Prime editing: An all-rounder for genome editing” by Lu, et al., talks about the potential of prime editing as a versatile genome editing technology as compared to the already well-established GE techniques.

This manuscript is well written and thoroughly described and it can find interest among the researchers in this field. However, there are some important corrections needed which must be incorporated before its possible publication.

Moreover, the language used in this manuscript needs serious attentions as well. There are some major/minor grammar and spelling and sentences errors throughout the manuscript that should be edited by some native English-speaking expert.

In the introduction section, there is a need to develop a bridge of arguments between base editing system and PE techniques to support the efficiency of PE technology with latest bibliography.

The figures are not clear and of poor quality. They should be replaced

There is a need to add some latest bibliography in a tabular description for the applications of PE systems in medicine and agriculture separately to list the already work done in this specific technique.

Say something about the drawbacks/shortcomings of this technique? Moreover, what are the effects of RNA stability on pegRNA efficiency?

What about the methods of application of PE systems?

How the use of different Cas proteins with increased PAM flexibility can play their role to overcome the limitations of the prime editing?

What is the role of PE technique for pathogenic mutations?

Authors should also compare targeted and off-targeted PE systems supporting with latest bibliography

To sum up, the MS appears generally well conceived and contains some new information.

Good Luck!

Reviewer 2 Report

The review article by Lu. C et al titled as “Prime editing: An all-rounder for genome editing” discusses the developments and applications of 'prime-editing', a genome editing technology. The authors did a great job of summarizing the recent developments made in the genome editing (PE) technology. The article is well-structured and comprehensive.

Minor corrections:

1.     Figures 1 and 2, labels/fonts can be bigger. 

2.     In table 1, in efficiency column, what does * indicate? 

3.      In line 65, …‘end recombination’ can be replaced with ‘end joining’.

Round 2

Reviewer 1 Report

Dear Authors, 

many thanks for your kind response!